# Back Injection Molding of Sub-Micron Scale Structures on Roll-to-Roll Extrusion Coated Films

**DOI:** 10.3390/polym13091410

**Published:** 2021-04-27

**Authors:** Sijia Xie, Jerome Werder, Helmut Schift

**Affiliations:** 1Laboratory for Micro- and Nanotechnology, Paul Scherrer Institute (PSI), 5232 Villigen, Switzerland; sijia.xie@psi.ch; 2Institute of Polymer Nanotechnology (INKA), FHNW University of Applied Sciences and Arts Northwestern Switzerland, 5210 Windisch, Switzerland; jerome.werder@fhnw.ch

**Keywords:** printed flexible electronics, back injection molding, roll-to-roll extrusion coating, conductive nanoparticle ink, thermal bonding

## Abstract

Roll-to-roll extrusion coated films were bonded onto polymer parts by back injection molding (BIM). The polypropylene (PP) coated polyethylene terephthalate (PET) films were pre-patterned with microstructured V-shaped grooves with 3.2 µm and 53 µm width, and other geometries. Bonding on PET and poly(methyl methacrylate) (PMMA) parts was facilitated by either higher tool or melt temperatures but was particularly enhanced by applying a mild oxygen plasma to the backside of the PET film prior to injection of the polymer melt. Silver wires from conductive nanoparticle ink were embedded into the PP coating during the BIM process by controlled collapse of the V-grooves. Thus, the feasibility of combining standard carrier film materials for printed flexible electronics and packaging into a non-flat polymer part was demonstrated, which could be a helpful step towards the fabrication of polymer parts with surface functionality.

## 1. Introduction

Back injection molding (BIM) is a mass manufacturing process for adding films with integrated functions onto the surface of bulk plastic products [1]. In addition to its traditional use in surface decorating, labeling, and laminating plastic products, BIM also offers a fast solution for integrating printed flexible electronics into three-dimensional polymer parts [2,3]. We have previously demonstrated a unique fabrication process that combines planar surface patterning with nanoimprint lithography (NIL) and BIM, which realizes sub-micron wide, millimeter long conductive metal wires on a cylinder-shaped surface [4]. In comparison to normal injection molding (IM), where the plastic part replicates the surface pattern from a patterned cavity surface, BIM enables independently patterning a flat polymer film (skin) and then bonding it to a plastic element (body) during the BIM process [5,6]. The hybrid part forms upon filling a mold cavity with a viscous melt, typically by using the same material for the film and as granulate for the melt. The melt and tool temperatures (T_melt_ and T_tool_) for the process have to be adapted to the glass transition temperature (T_g_) of the film, in a way in which the process leads to an intimate contact and sufficient bonding while maintaining the surface structures of the film. With a T_melt_ higher than the melting temperature T_m_ of the injected polymer and T_tool_ lower than its T_g_, the injection process is performed in an isothermal manner. However, the process is highly dynamic, because the film is softened by the melt and cools rapidly upon contact with the mold wall. We performed initial research with an amorphous thermoplastic polymer, poly(methyl methacrylate) (PMMA) as both film and melt. Good bonding conditions can be found for PMMA, when materials with similar T_g_ are used. However, further investigation is needed on materials such as polyethylene terephthalate (PET), which is conventionally used as substrate for flexible electronics. In this case, the melt for constituting the polymer element can be either PMMA, as in the previous research, or PET. The specific challenge is then to find good bonding conditions, since as a semi-crystalline polymer, PET does not have a pronounced T_g_, and therefore the interdiffusion of polymer chains during bonding is limited in comparison to the amorphous PMMA.

As an extension to our previous work focusing on PMMA, we demonstrated the combination of roll-to-roll extrusion coating (R2R-EC) and BIM to fabricate micro- and nanostructured double-layer polymer films on non-flat surfaces and compared it to the results obtained with NIL. R2R-EC is common in the packaging industry and widely used for the production of smooth polymer films [7,8]. In R2R-EC, a molten polymer film is extruded through a flat nozzle, then stretched and finally laminated onto a carrier film (substrate). The lamination process takes place as the melt curtain is squeezed between a structured cooling roller and a rubber counter roller (Figure 1a). This results in a double-layer polymer film in which the structured coating bonds to the carrier film. In comparison to films with thermally imprinted surfaces, which involve heating and cooling of a solid film, roll-to-roll (R2R) NIL processes facilitate the patterning of surface topography, since the molding happens upon contact of the melt with the cool structured roll, leading to immediate solidification, and thus providing high throughput in a continuous process.

PET is one of the typical materials for flexible electronics, owning to its high heat stability, solvent resistance, and good electrical properties. Polypropylene (PP) is a low-density thermoplastic material, which makes it a good candidate for the molding or extrusion of plastic parts with a lower cost for industrial applications, e.g., food packaging. In addition, its high resistance to acid and alkaline-based solvent makes it a good option for ink printing and painting. These materials are highly interesting for the flexible electronics industry. However, unlike PMMA, PET and PP are partially crystalline materials and do not show strong adhesion properties with polymer melts, e.g., PET film with PET or PMMA melt. Surface activation by gas plasma treatment [9,10,11] can be introduced prior to the BIM process to strengthen the bonding quality. By creating reactive oxygen components on the PET film surface we achieved strong bonding between the PET film and structured coating from both, PET and PMMA melts. The plasma treatment can be completed within 1 min, resulting in a purely chemical activation, which is efficient within at least 2 h in air.

The new approach here is to use pre-structured PP/PET films, which are fabricated by the R2R-EC method. Our results confirmed the viability of using different thermoplastic materials for mass fabrication of three-dimensional molded products, which have potential application perspectives in fields such as optics and flexible electronics. Here, we do not show a complete device, but rather show how silver nanoparticle-based wires could be integrated into a film.

## 2. Materials and Methods

The inserted PP/PET films were prepared using R2R-EC by Danapak Flexibles A/S, Slagelse, Denmark, via InMold Biosystems A/S, Hørsholm, Denmark, which provided unstructured, as well as structured, films with different patterns such as V- and square-shaped grooves. The typical thickness of the PET carrier film is ~50 µm and the laminated PP coating is in the range of 10 µm to 50 µm, depending on the process parameters such as speed, extrusion rate and as pre-adjustment of the gap, and the structures on the roll [7,8]. Borealis Daploy™ WF420HMS high melt strength long-chain branched propylene homopolymer was used for extrusion coating. It has a melting temperature T_m_ of 162–165 °C (ISO 3146) and a Vicat softening temperature (VST) of 95 °C (B/50 (50 N); ISO 306). Although this value is higher than the T_g_ of 73–78 °C given for PET, it is still well below the melting temperature of PET. As a semicrystalline polymer, PET is not fully softened at this temperature. Therefore, PP can be thermally imprinted at 160 °C without deformation of the PET. Detailed information of the PP/PET films used in the experiment is included in Table 1.

In brief, for R2R-EC, polymer granulate (here PP) was melted and injected between the carrier film (here PET) on a supporting roll (roll 1) and a cooling roll (roll 2), which was kept at a temperature lower than the solidification temperature of the polymer (Figure 1a). Once coming into contact with both roll 2 and the carrier film, the melt cools down while being co-extruded with the carrier foil, driven by the rotating rolls (roll 1, 2, and 3). The polymer melt solidifies and adheres on the carrier film, and detaches from roll 2, forming a two-layer film. If roll 2 includes a mold with structures on the surface, the structures will be replicated in the melt layer. A flat roll 2 will result in a flat film. In our experiments, a series of structures fabricated by R2R-EC were used. Detailed information of the films and the structures are listed in Table 2.

Additionally, we patterned flat PP/PET films with a 4 µm period and 2.1 µm deep V-grooves by thermal NIL, similarly as we did with the PMMA films in the previous work (Figure 1b). Details of the fabrication of the working stamp in Ormostamp was introduced in [4]. Here, the PP/PET films were thermally imprinted in the PP layer for 10 min at 160 °C, employing a pressure of 2.5 MPa, followed by cooling and subsequent demolding at 80 °C in a Jenoptik HEX03 thermal imprint machine. The exact thickness of the films was not indicated by the suppliers (InMold, Danapak) [12].

We used a silver nanoparticle-based ink (Smart’Ink S-CS31506, Genes’Ink, Rousset, France) to generate silver wires on the pre-patterned polymer films. The original ink was a suspension of silver nanoparticles in alcohol and glycol (nanoparticle loading of 55 *w*/*w*%), which was diluted with isopropyl alcohol (IPA) at a volume ratio of 1:59, resulting in a nanoparticle concentration of 0.17 *v*/*v*%. This solution was spin coated on the patterned samples for 120 s at 5000 rpm. The diluted ink covers the surface with a low density, so that no conductive film is created, due to the sparse distribution of particles. However, in our case, the geometry of the V-grooves helped to confine the particles during spin-coating (Figure 1c), therefore thin wires were created by aggregation upon drying. More details are presented in [13,14,15]. As a demonstration of the film deformation, here the spin coated silver nanoparticles were not thermally sintered to achieve the optimized conductive property.

In order to improve the bonding quality between the PP/PET film and the granulate, we treated some of the PET films on the PET side with a mild oxygen plasma treatment prior to the BIM experiment. The plasma treatment was performed in a Plasmalab 80 Plus (Oxford Instruments plc, Abingdon, UK). The optimal treatment parameters for this tool are 20 sccm O_2_, 20 mTorr, and 40 W RF power for 40 s, which were used on all the samples discussed in the Results and Discussion section.

We performed the BIM in an Arburg 320 Allrounder (Arburg, Lossburg, Germany). The maximum clamping force of the machine is 600 kN. A modular injection molding tool with a cylinder-shaped curved cavity was fixed in the three-plate molding tool, a HASCO “handy mold” with ejector pins. The pre-patterned polymer film was inserted on the side of the concave cavity; therefore, it was be integrated onto the convex surface of the molded part. A polyimide film (Kapton^®^ 200 HN from DuPont, Paris, France, 50 µm thick) was placed between the inserted film and the metal tool wall to minimize the surface roughness of the molded part caused by the metal. During the BIM process, polymer granulate (PMMA or PET) was heated to an optimized T_melt_ > T_m_, typically 250 °C for PMMA, and 270 °C for PET. Subsequently, the melt was injected into the cylinder tool, which had a much lower temperature. T_tool_ was controlled by circulating water. Upon contact with the tool and the inserted polymer film, the polymer melt cooled down and solidified, resulting in a 3 mm thick curved polymer element with the inserted film integrated into the convex surface of the element. A schematic of the BIM process is illustrated in Figure 2. Molding trials were conducted at a series of constant T_tools_ of 40, 50, and 60 °C. Detailed information of the polymers used in the BIM experiment can be found in Table 1.

We then examined the samples with a scanning electron microscope (SEM) (Supra VP55 from Zeiss, Oberkochen, Germany) at an acceleration voltage of 0.5–1.0 kV. For cross-sectional SEM imaging, we used different methods according to the material and condition of the samples. For flat PMMA film, the cross-section was made by immersing the film in liquid nitrogen for 3 min, and then immediately breaking it with a flat-end tweezer. For PMMA films bonded on the BIM part, the entire BIM part needed to be broken to obtain a cleaved facet at the desired location of the film. First, a notch was sawn into the concave backside of the element, then the film was broken at room temperature. The PP/PET films were too elastic to break with the aforementioned methods, therefore, a cross-section was created by first peeling off the weakly bonded PP/PET films (without plasma treatment before BIM) from the polymer element, then cutting the film at the desired location with a surgical blade.

## 3. Results and Discussion

### 3.1. Films with 4 µm Period V-Grooves by NIL

In order to compare the deformation behavior of the different inserted polymer film materials, we imprinted the same V-grooves (period 4 µm, groove depth 2.1 µm, groove opening width 3.2 µm) into the flat PP layer of the PET film as described above. Similarly to our previous results for PMMA, the V-grooves in the PP layer underwent a squeezing-like deformation after the BIM process, which can primarily be described as a collapse of the sidewalls into a 1 µm deep trench with almost vertical sidewalls. Therefore, as shown before [4], metal agglomerates at the bottom of a V-groove would stay at the bottom of that trench, making it possible to embed metal wires below the surface of a completely flattened surface. Figure 3 (left column) shows the deformation of the V-grooves in the PP layers, with a T_tool_ of 40, 50, and 60 °C, respectively. The sidewalls of the V-grooves already collapse at a T_tool_ of 40 °C, resulting in a narrower and shallower trench, and a widened plateau between the trenches. However, as the T_tool_ rises, the profile of the V-grooves does not change much. In contrast, the V-grooves in the PMMA film did not deform as strongly as in the PP layer at 40 °C, but they seemed to be more sensitive to the temperature change (Figure 3, right column). In the case of the PP/PET film used in Figure 3, the total thickness was around 100 µm including the PET carrier film. The PMMA film was 175 µm thick. The Young’s modulus of PP is in the range of 1.5–2 GPa, while that of PMMA is 2.4–3.4 GPa at room temperature [16]. The thermal conductivity of PP, PET, and PMMA are all within the range of 0.1–0.2 W/mK, therefore, the thermal transmission of the film is mainly inversely proportional to the film thickness. Given the lower mechanical properties, as well as the thinner film thickness, it is then logical that the structure in the PP/PET film deforms more strongly than in the PMMA at 40 °C, as PP is softer than PMMA at low temperatures, and the temperature of the film surface is supposed to be higher when it contacts the tool surface. Although PP has a lower VST, as an amorphous material, PMMA is more sensitive to temperature variations than the semi-crystalline PP.

### 3.2. Metal Wire Sealing on Different Surface Structure Dimensions

Previously, we investigated the possibility of embedding metal wires that were first spin-coated on a flat film, and then onto the cylinder-shaped BIM part (similar to the product shown in Figure 6), using PMMA as both the film and granulate material for IM. For PMMA, we were able to embed 600 nm wide silver wires at approx. 1 µm depth from the surface with BIM. In the case of imprinted V-grooves in PP/PET film, a similar behavior can be observed in Figure 4a,b at a T_tool_ of 40 °C. However, in order to seal the trench opening on the surface completely a higher tool temperature is needed for PP, since the deformation does not vary from 40 to 60 °C, as discussed above in the T_tool_ series experiment. In addition to the effect of the temperature condition, the geometry of the pre-patterned structures also plays an important role in the wire fabrication. For example, the width of the wire, here formed by the spin-coated silver nanoparticles, varies according to the dimensions or the cavity volume of the pre-patterned V-grooves prior to BIM. We spin-coated silver ink on larger V-grooves (period 60 µm, groove depth 28 µm) in the PP/PET film, which were fabricated by R2R-EC. In the large V-grooves, 3.8 µm wide silver particle wires were formed (Figure 4c), while the same concentration of the silver ink and spin-coating recipe are used for both the small and the large V-grooves. On the other hand, the ratio of the wire width (W_w_) to the groove dimension (D_g_, including width, depth) is a determining parameter for the wire embedding. The lower this ratio is, the easier the wire is embedded. Taking T_tool_ = 40 °C as an example, the deformation in the large V-grooves is different from that of the small V-grooves with 4 µm period and 2.1 µm depth. The large V-grooves are already almost fully flattened out at 40 °C, leaving an approx. 70 nm wide groove opening on the surface (Figure 4d).

The deformation of the structures creates various strategies for fabricating metal wires that are pre-processed on a flat substrate. With a low ratio of W_w_/D_g_, complete wire embedding can be realized even at a low T_tool_ for BIM. This would avoid the potential damage of using a high T_tool_ that might result in softening of the polymer film and hence damage or even discontinuity in the wires. A high W_w_/D_g_ value, on the other hand, could be utilized to remove the pre-patterned structures by the fast BIM process, instead of thermal reflow for a long time [15], leaving the wires exposed on the surface for further processes.

### 3.3. PET Film Bonding with PET and PMMA Melt in BIM

In order to learn more about the relationship between structure dimension and structure deformation in BIM, we also evaluated PP/PET films with different pre-patterned square grooves in a PP layer fabricated by R2R-EC. We observed two types of deformation in the square grooves with vertical sidewalls: (1) The grooves flatten out after BIM (Figure 5a–c), while the trace of the sidewalls remains vertical in the cross-sectional view. In this case, the bottom of the grooves seems to be “pushed” towards the surface, along the direction of the injection. The groove width does not change after BIM. This deformation mainly happens in structures with large groove width. (2) The vertical sidewalls collapse during the flattening of the structures, resulting in a narrowed groove width and a triangle/ trapezoid shape-undercut at the cross section (Figure 5d–h). In Figure 5g,h, the sidewalls are shielded by the distorted PP layer. This was caused by blade cutting during the preparation of the cross-sections, which gives a false appearance that the sidewalls remain vertical. The ratios of groove width to groove depth in Figure 5d–h are 0.8, 2.5, 1.65, 1.27 and 1, respectively. As can be seen from the top surface of the films after BIM in Figure 5, a higher structure aspect ratio results in stronger collapsing of the sidewalls, and therefore a narrower groove opening after BIM. This is logical, as with high aspect ratio structures, more material is compressed in a vertical direction and then pushed along the horizontal direction. Based on the dimensions we investigated, the aspect ratio does not seem to determine which type of deformation the structure will undergo during BIM. For example, the films in Figure 5b and h have the same aspect ratio of 2, however, the deformation falls into two types. The deformation mechanism is a result of the combined effects of the geometry and dimensions, as well as the softness of the film when the polymer melt pushes it into contact with the mold cavity wall. We performed a simulation of the deformation of the films in Appendix A. For case (1), it could be that the large-scale opening provides sufficient space in the horizontal direction for the softened polymer to be squeezed into the cavity, and this squeezing process happens before the ridge can deform. Meanwhile, for case (2), the small dimension of the groove opening, which can be considered as a secondary “mold” cavity for the polymer at the bottom, hinders the filling of the polymer, while the compression of the structures as a whole has a dominant role during the deformation, leading to collapsing sidewalls. Dedicated melt flow simulation would help to better understand the deformation process, which is not the focus of this paper.

### 3.4. PET Film Bonding with PET and PMMA Melt in BIM

In general, the PMMA film bonds well with the PMMA melt for a wide range of T_tool_ (40–80 °C) when the grade of the film and granulate are similar. Unlike PMMA, the PET film does not bond strongly with the solidified PET melt, mostly because of its semi-crystalline conformation, which restricts interdiffusion of the polymer chains. At a T_tool_ of 40 °C, the PET film bonds weakly with the solidified PET melt, and the bonding is enhanced as the T_tool_ increases. However, the film can still be peeled off from the surface with a gentle force, e.g., by hand. Surface activation, such as oxygen plasma treatment on the PET film surface, significantly strengthens the bonding between the film and the solidified melt. Already with a mild activation, the bonded PET films cannot be detached without damaging the films. We made a hand-operated pull-off test to evaluate the pull-off force of the bonded film. In the cases of both PET/PET and PET/PMMA bonding, the pull-off force was <1 N for non-treated films, and >10 N for plasma-treated films, where the films were damaged in the latter case. The setup and results of the hand-operated pull-off test are displayed in Appendix A and Table 3, respectively. Figure 6a shows the improvement of the bonding between PET films after an optimized oxygen plasma treatment. Therefore, a longer oxygen plasma treatment, which has been found to result in surface roughening in different polymers, is not needed for further improvement [17].

We also tested hybrid bonding, i.e., a PET film with solidified PMMA melt. The untreated PET film did not bond with PMMA at a T_tool_ of 40 °C. However, after plasma surface activation, the PET film bonded with PMMA strongly enough to avoid delamination (Figure 6b).

The bonding quality between the polymer film and the polymer melt depends on the surface chemistry of the material, as well as the contact temperature between the film and the melt. PET is a relative inert material with low surface free energy, and high thermal stability and chemical resistance, which makes it a poor candidate for surface adhesives [18]. Therefore, bonding between PETs is supposed to be difficult to achieve without melting the PET, at least at the interface of each of the bonding parts. This explains why the PET film did not bond strongly with the PET melt at a low T_tool_ range (40–60 °C) in our experiments. An oxygen plasma treatment on the surface activates the PET by generating more reactive hydroxyl groups on the surface, hence making it possible to chemically bond to the molecule chain of the PET melt [10,11]. In the case of the PMMA melt, the bonding between the activated PET and the PMMA melt is even stronger at the same melt temperature, owing to the higher viscosity of the PMMA, which allows interdiffusion of chains near the interface, as well as the free sidechains in the PMMA molecules.

The deformation and bonding behavior of the film in the BIM process mostly depends on the properties of the polymer material. Another dominant experimental condition is the contact temperature between the film and the melt. Unfortunately, so far this value has not been possible to directly measure, but only be simulated according to the material type and grade, film thickness, and the experiment parameters of the BIM machine. As demonstrated in Table 3, the bonding could be improved by using a higher melt temperature T_melt_ and a higher tool temperature T_tool_. The temperature at the interface is in the range between these two temperatures. Once the film is pressed onto the tool wall, the melt is rapidly cooled and solidifies. Then, both the structure collapse and the film bonding happen. Among the structures tested in our experiment, structures with a high aspect ratio and narrow groove opening underwent a sidewall collapsing process stronger than flattening out, and hence are better for enabling the embedding of pre-processed silver wires. We could learn and investigate an optimized process flow through tuning the experiment conditions accordingly and characterizing the resulting products, however, the process optimization would definitely benefit from a customized simulation.

## 4. Conclusions

The experiments demonstrated that the good BIM results already obtained with PMMA films on a solidified PMMA melt can be transferred to PET. This makes it possible to combine the standard carrier material for printed flexible electronics with parts made from PMMA and PET melts. In addition to the bonding behavior during BIM, the embedding of wires using a defined collapse of V-grooves was shown for a two-layer system, where V-grooves were imprinted into a R2R-EC patterned PP film. This would enable combining two processes that could be upscaled to high volume fabrication. Even the generation of metallic microstructures from nanoparticle solutions and an additional surface treatment step before BIM would not add much complexity to the whole process. Although PP is not a material commonly used in printed flexible electronics, it is highly resistant against chemical attacks and, by embedding sub-micron metal wire arrays and meshes, could be used for electrostatic shielding in packaging, housings, or containers. In principle, other thermoplastic polymer materials should also work similarly under adapted experimental conditions. By demonstrating the combination of two state-of-the-art manufacturing processes, the use of more complex geometries and polymers seem to be feasible for integrating micro to nanoscale structures and electrically conductive wires onto non-flat surfaces.

## Figures and Tables

**Figure 1 polymers-13-01410-f001:**
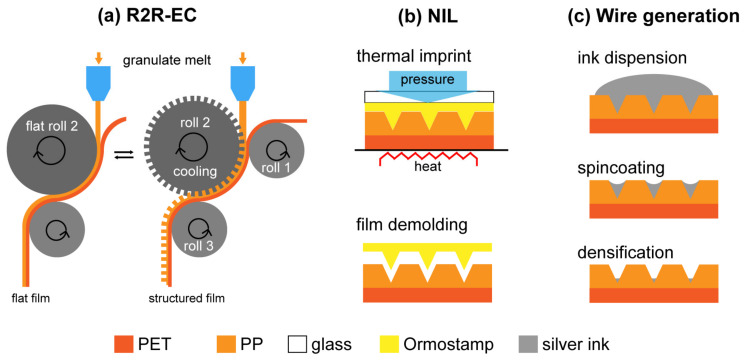
Fabrication processes of the polymer films. (**a**) Roll-to-roll extrusion coating (R2R-EC): An unstructured, polished cooling roll 2 (left side) results in a flat PP layer, while a roll 2 with surface structures (right side) results in structured PP layer on a PET carrier film. (**b**) Nanoimprint lithography (NIL): The flat PP/PET film is imprinted with a structured Ormostamp mold under heat and pressure. After demolding, V-groove structures are present in the PP/PET film. (**c**) Wire generation: By spin-coating silver nanoparticle ink on the pre-patterned PP/PET film, silver wires are generated by confinement of the structures during solvent evaporation.

**Figure 2 polymers-13-01410-f002:**
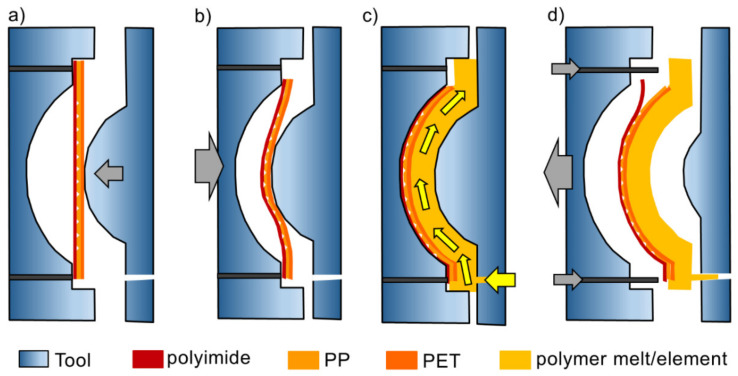
Schematic of the BIM process. (**a**) Insertion of patterned film, fixed at bottom of concave side of the molding tool; (**b**) Closing of mold cavity, pre-bending of film; (**c**) Injection of melt from nozzle (injection gate) at the bottom of the right side of the molding tool; (**d**) Demolding of polymer part with ejector pins from the left side. Adapted from ref. [4].

**Figure 3 polymers-13-01410-f003:**
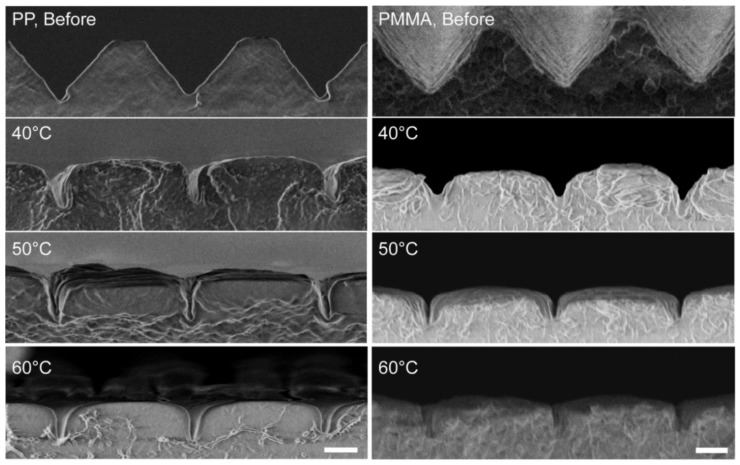
Comparison between imprinted 100 µm thick PP/PET (left column) and 175 µm thick PMMA (right column) films, before (top row) and after the BIM process: In both cases, the sidewalls of the original (empty) V-groove structure with 4 µm period and 2.1 µm groove depth collapses into a trench with ~1 µm. Scale bar: 1 µm. The SEM images of PMMA are adapted from Figure 3 of [4].

**Figure 4 polymers-13-01410-f004:**
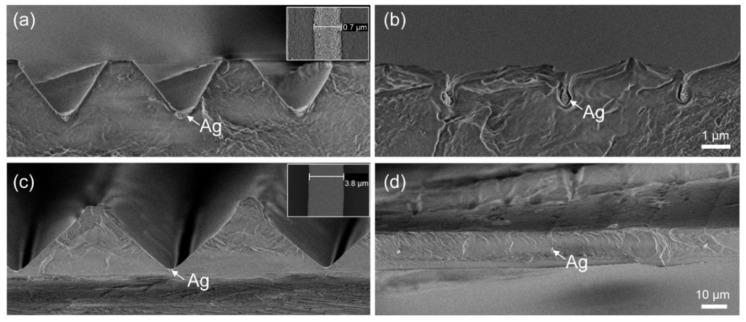
Comparison of embedded silver nanoparticles on different surface structures before (**a**,**c**) and after (**b**,**d**) BIM. (**a**,**b**) Small V-grooves with 4 µm period. Scale bar: 1 µm. (**c**,**d**) Large V-grooves with 60 µm period. Scale bar: 10 µm.

**Figure 5 polymers-13-01410-f005:**
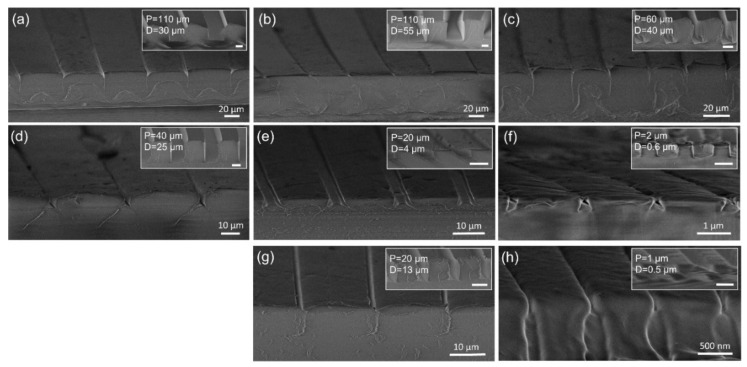
Deformation of the square grooves in PP after the BIM process. (**a**–**h**) Cross-sectional profile imaged in SEM. The original structures and their dimensions are displayed on the upright corner of each image.

**Figure 6 polymers-13-01410-f006:**
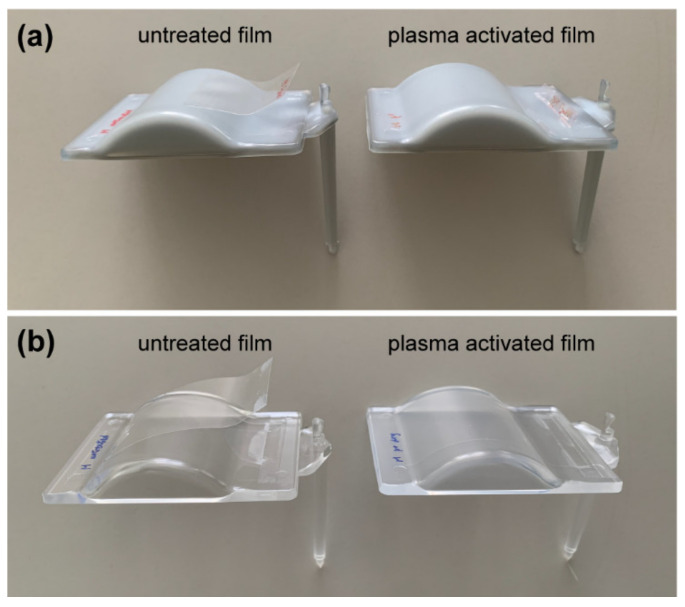
Bonding between a PET film and a PET element (**a**) or PMMA element (**b**), without and with oxygen plasma surface activation. T_melt_ was 270 °C for PET granulate and 250 °C for PMMA granulate, and T_tool_ was 40 °C. With activation, the bonding is strong enough that the film cannot be delaminated without damage.

**Table 1 polymers-13-01410-t001:** Comparison of the polymers used as films for R2R-EC (PP on PET) and as granulates for BIM (PMMA and PET).

Material	Supplier	Glass Transition Temp. T_g_	Vicat Softening Temp. VST (B/50)	Melting Temp. T_m_	Recommended Melt Temp. T_melt_ for IM
PMMA film 175 µm	Evonik Plexiglas^®^ film 99,524	113 °C	100 °C	n/a ^1^	n/a
PP granulate/film	Borealis Daploy™ WF420HMS	−20–0 °C	95 °C ^2^	162–165 °C	130–170 °C
PET carrier film	n/a	73–78 °C ^3^	n/a	~260 °C ^4^	n/a
PMMA granulate	Evonik Plexiglas^®^ 7N	110 °C	103 °C	~160 °C ^1^	220–260 °C
PET granulate	Saxaplast PET-01 BTI K—N001	n/a	75 °C	n/a	250–270 °C

^1^ As an amorphous material, PMMA does not exhibit a pronounced T_m_. ^2^ Data from Borealis Polypropylene BE50. ^3,4^ Values for general PET material.

**Table 2 polymers-13-01410-t002:** Information of the polymer films used in the experiment.

Film Material	Patterning Method	Structure	Period (µm)	Groove Width (µm)	Groove Depth (µm)	Total Film Thickness ^1^ (µm)
PMMA	NIL	V-groove	4	3.2	2.1	175
PP/PET	NIL	V-groove	4	3.2	2.1	100
	R2R-EC	V-groove	60	53	28	80
		Square groove with vertical sidewalls	110	55	30	90
			110	55	55	110
			60	30	40	100
			40	20	25	90
			20	10	4	90
			2	1	0.6	60
			20	10	13	90
			1	0.5	0.5	60

^1^ Total thicknesses were measured by a caliper and cross-sectional SEM.

**Table 3 polymers-13-01410-t003:** Pull-off test force of the bonded film-body. The high peeling forces in samples 7, 9, 10, and 12–14 indicate a good attachment between film and element, since the films could not be detached without breaking.

Sample	Film (Skin)	Element (Body)	Surface Treatment	Melt Temp. (°C)	Tool Temp. (°C)	Peeling Force (N)	Film Condition during Test
1	PET	PET	no	270	40	0.490	peeled
2	PET	PET	no	270	40	0.588	peeled
3	PET	PET	no	270	40	0.686	peeled
4	PET	PET	no	270	40	0	instantly detached
5	PET	PET	no	270	40	0	instantly detached
6	PET	PET	plasma	270	40	16.2	PP layer delaminated
7	PET	PET	plasma	270	40	23.6	broken from edge
8	PET	PMMA	no	250	40	0	instantly detached
9	PET	PMMA	plasma	250	40	21.4	broken from edge
10	PET	PMMA	plasma	250	40	15.2	broken from edge
11	PET	PMMA	plasma	260	30	5.6	peeled
12	PET	PMMA	plasma	260	30	8.6	broken from edge
13	PET	PMMA	plasma	260	30	29.2	broken from edge
14	PET	PMMA	plasma	260	40	23.6	broken from edge
15	PET	PMMA	plasma	260	40	24.2	PP layer delaminated

## Data Availability

Not applicable.

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
