# Peer review of "Back Injection Molding of Sub-Micron Scale Structures on Roll-to-Roll Extrusion Coated Films"

_polymers, 2021, doi:10.3390/polym13091410_

Round 1
Reviewer 1 Report
Well written, but there are some improvements which should be performed:
Line 65 and 78: thermal plastic CORRECT: thermoplastic
Line 89: "Borealis Daploy™ WF420HMS high melt strength propylene-based, structurally isomeric polymer for extrusion coating" COMMENT: I understand this is taken from a general producer's description, but I recommend an understandable form: Borealis Daploy™ WF420HMS high melt strength long-chain branched propylene homopolymer for extrusion coating.
Line 92
Line 92 Although this value is higher 92 than the Tg of 73 – 78°C given for PET, the PP can be thermally imprinted at 160°C without deformation of the PET COMMENT: PET is a semicrystalline polymer which does not fully soften at its glass temperature which refers to the amorphous phase.
Table 1: QUESTION: Why the supplier of PET carrier film is not known?
Line 322: "With PET, at temperatures below Tg where crystallites are present in the film, the bonding seems to be less dependent on chain movement"
COMMENT: In PET crystallites are present below and above its glass temperature until the material melts. Below Tg, there is no chain movement in the amorphous phase.
Author Response
Dear Editor,
We would like to thank the reviewers for their thorough evaluation of our manuscript, and for their comments that bring a clear improvement on the manuscript. We have carefully read the comments and have made corrections & changes accordingly. Please find enclosed a marked-up version where all the changes are marked in highlight (yellow), as well as a clean version.
With best regards,
Helmut Schift
Line 65 and 78: thermal plastic CORRECT: thermoplastic
We thank the reviewer for pointing out this mistake. We have corrected the term to “thermoplastic” in Line 65 and 78.
Line 89: "Borealis Daploy™ WF420HMS high melt strength propylene-based, structurally isomeric polymer for extrusion coating"
COMMENT: I understand this is taken from a general producer's description, but I recommend an understandable form: Borealis Daploy™ WF420HMS high melt strength long-chain branched propylene homopolymer for extrusion coating.
We have changed the sentence in Line 89 to “Borealis Daploy™ WF420HMS high melt strength long-chain branched propylene homopolymer was used for extrusion coating. It has a melting temperature…”.
Line 92 Although this value is higher 92 than the Tg of 73 – 78°C given for PET, the PP can be thermally imprinted at 160°C without deformation of the PET
COMMENT: PET is a semicrystalline polymer which does not fully soften at its glass temperature which refers to the amorphous phase.
We thank the reviewer’s comment on making the explanation accurate and more clearly. We have changed the quoted sentence in Line 92~95 to “Although this value is higher than the Tg of 73 – 78°C given for PET, it is still well below the melting temperature of the PET. As a semicrystalline polymer, the PET is not fully softened at this temperature. Therefore, the PP can be thermally imprinted at 160°C without deformation of the PET.”
Table 1: QUESTION: Why the supplier of PET carrier film is not known?
We have asked the company Danapak for providing this information but do not have an answer yet. This could probably be a sensitive information. In any case, we expect the the PET film is a standard material and that the process described in this paper will be general enough that the results are valid. In case we get this information within the next days, we would like to include this before the "proofs" are accepted.
Line 322: "With PET, at temperatures below Tg where crystallites are present in the film, the bonding seems to be less dependent on chain movement"
COMMENT: In PET crystallites are present below and above its glass temperature until the material melts. Below Tg, there is no chain movement in the amorphous phase.
We thank the reviewer for correcting our inaccurate explanation for the PET chemistry. We have reviewed the paragraph, and have decided to remove the quoted sentence in Line 322 (Line 327 in the revision. We have moved Table S1 to the main text as Table 3, as suggested by another reviewer). We think that the first half of the paragraph has provided enough information about the bonding difficulties between PET film and PET granulate, and the discussion about the chain movement seems to add confusion rather than clarity.
Reviewer 2 Report
This paper focuses on the feasibility of a new process to fabricate the polymer parts with surface functionality. In general, the experiments are well designed and the results and conclusions presented are, most of the time, clearly explained. I think it can be acceptable for publication but need some revisions.
1-please provide plasma treatment parameters.
2-Pull off test results should put in the manuscript, which are very important to evaluate feasibility of the combined process proposed in this work.
Author Response
Dear Editor,
We would like to thank the reviewers for their thorough evaluation of our manuscript, and for their comments that bring a clear improvement on the manuscript. We have carefully read the comments and have made corrections & changes accordingly. Please find enclosed a marked-up version where all the changes are marked in highlight (yellow), as well as a clean version.
With best regards,
Helmut Schift
Author's Reply to the Review Report (Reviewer 2)
Comments and Suggestions for Authors
This paper focuses on the feasibility of a new process to fabricate the polymer parts with surface functionality. In general, the experiments are well designed and the results and conclusions presented are, most of the time, clearly explained. I think it can be acceptable for publication but need some revisions.
1-please provide plasma treatment parameters.
We thank the reviewer’s remark on the missing information. We have added the detailed plasma treatment parameters in Line 142 to 147:
In order to improve the bonding quality between the PP/PET film and the granulate, we treated some of the PET films on the PET side with a mild oxygen plasma treatment before the BIM experiment. The plasma treatment was performed in a Plasmalab 80 Plus (Oxford Instuments). The optimal treatment parameters for this tool are 20 sccm O2, 20 mTorr, 40 W RF power for 40s, which were used on all the samples discussed in the Results and Discussion section.
2-Pull off test results should put in the manuscript, which are very important to evaluate feasibility of the combined process proposed in this work.
As suggested, we have included the results (Table S1) of the pull off test in the main text of the manuscript, as Table 3. We keep the photo of the pull off test setup (Figure S2) in the Supplementary Information; however, we are open to additional suggestions.
In Line 300, we have changed the “Table S1” to “Table 3”.
In Line 313, we have added the pull off results, moving from Table S1 to Table 3.
This manuscript is a resubmission of an earlier submission. The following is a list of the peer review reports and author responses from that submission.
Round 1
Reviewer 1 Report
The article is well written. presents an interesting fabrication method with a perspective of implementation in the praxis. For the future I advise to measure the effect of plasma treatment (FTIR estimation of carbonyl and hydroxyl groups) and the bonding strength, to define the conditions and requirements for these operations.
I noticed small mistakes:
Line 61: Polypropylene (PP) is a low-density thermal plastic material - CORRECT : thermoplastic material
Line 63: In addition, its high resistance to acid and alkaline- based solvent makes it a good option for ink printing and painting.
REMARK: Not true, untreated polypropylene has low printablity
"Untreated polyolefin-based polymers have a number of undesirable characteristics including poor wettability, poor printability, and poor adhesion to secondary phases. These properties are associated with the low surface energy of polyolefin-based polymers and their high resistance to most chemicals and solvents. "( Surface properties and printability of air dielectric barrier discharge plasma-treated polypropylene film, Surface Coatings International Part B: Coatings Transactions 1 Vol.88, Bx, xx–xxx, ? 2005)
Line 73: different thermal plastic materials - CORRECT: thermoplastic materials
Reviewer 2 Report
The paper is generally well written, but there are some issues.
1) There is little novelty in this work. First, results from the processes have previously been reported for one polymer. Second, the softening and associated issues with PP compared to PMMA have previously been reported for a wide range of processes. Third, it is not clear that PP would be suitable for flexible electronics. Fourth, the issue of sintering the Ag and its relationship to the polymer systems was not addressed. [Examination of a more polymers or polymers suitable for flexible electronics would have been a more suitable follow up for the work with PMMA.]
2) The discussion of the materials could have a little clearer. Lines 78-87 = no discussion of the PMMA listed in Table 1. Lines 88-96 = was the PET pretreated to achieve good adhesion between the PET and PP melt? Line 98 = why were Tg and Tm for the PET granulate not reported in Table 1?
3) There are some grammar issues. Please check the verb tenses; there is a random use of past and present tenses. Please check for "thermal plastic" - which should be "thermoplastic" (e.g., lines 61, 73). Lines 65-66 = should be “PET and PP are semi-crystalline materials”.
Reviewer 3 Report
Nanomaterials “considers all original research manuscripts provided that the work reports scientifically sound experiments and provides a substantial amount of new information. Authors should not unnecessarily divide their work into several related manuscripts”.
However, this paper is largely based on other recent papers by the same authors, especially the followings:
[4] Xie, S.; Horváth, B.; Werder, J.; Schift, H. Sub-micron silver wires on non-planar polymer substrates fabricated by thermal nanoimprint and back injection molding, Micro and Nano Eng. 2020, 8, 100062.
[13] Horváth, B.; Křivová, B.; Bolat, S.; Schift, H. Fabrication of large area sub‐200 nm conducting electrode arrays by self‐confinement of spincoated metal nanoparticle inks, Adv. Mat. Technol. 2019, 3(4), 1800652 (11 pp).
With respect to the previous papers, the amount of new information is not substantial and is still in a preliminary form:
- In the production of films with V-grooves the same process proposed in [4] was employed. The authors represented the same results obtained in [4] for PMMA and added very similar results for PP. Even some figures are the same in both articles.
- The metal wire sealing in V-grooves was already published in previous papers. The results for PP are like the previous ones. The deformation of the structures fabricated by R2R-EC is described only qualitatively without attempting any modelling.
- Film bonding strength is neither measured quantitatively nor modeled. These results are very preliminary.